# Factors affecting decision-making in children with complex care needs: a consensus approach to develop best practice in a UK children's hospital

Giles Birchley ![ORCID],[1] Sadie Thomas-Unsworth,[2] Charlotte Mellor,[3] Mai Baquedano,[4] Susanne Ingle,[1] James Fraser[5,6]

¹Population Health Sciences, University of Bristol Medical School, Bristol, UK
²Clinical Psychology, Bristol Royal Hospital for Children, Bristol, UK
³Paediatric Palliative Care and Bereavement Support, Bristol Royal Hospital for Children, Bristol, UK
⁴Translational Health Sciences, University of Bristol Medical School, Bristol, UK
⁵Paediatric Intensive Care Unit, Bristol Royal Hospital for Children, Bristol, UK
⁶Paediatric Critical Care Society, London, UK

**Correspondence to**
Dr Giles Birchley; giles.
birchley@bristol.ac.uk

## ABSTRACT

**Background** Children with complex care needs are a growing proportion of the sick children seen in all healthcare settings in the UK. Complex care needs place demands on parents and professionals who often require many different healthcare teams to work together. Care can be both materially and logistically difficult to manage, causing friction with parents. These difficulties may be reduced if common best practice standards and approaches can be developed in this area.

**Objective** To develop a consensus approach to the management of complexity among healthcare professionals, we used a modified Delphi process. The process consisted of a meeting of clinical leaders to develop candidate statements, followed by two survey rounds open to all professionals in a UK children's hospital to measure and establish consensus recommendations.

**Results** Ninety-nine professionals completed both rounds of the survey, 69 statements were agreed. These pertained to seven thematic areas: standardised approaches to communicating with families; processes for interprofessional communication; processes for shared decision-making in the child's best interests; role of the multidisciplinary team; managing professional–parental disagreement and conflict; the role of clinical psychologists; and staff support. Overall, the level of consensus was high, ranging from agreement to strong agreement.

**Conclusions** These statements provide a consensus basis that can inform standardised approaches to the management of complexity. Such approaches may decrease friction between parents, children and healthcare professionals.

## INTRODUCTION

Children's complex care needs have been defined as 'multidimensional health and social care needs in the presence of a recognized medical condition or where there is no unifying diagnosis. They are individual and contextualized, are continuing and dynamic, and are present across a range of settings, impacted by healthcare structure'.[1] Many children with complex care needs also have life-limiting conditions, that is, a condition

### WHAT IS ALREADY KNOWN ON THIS TOPIC

⇒ The number of children with complex care needs in UK hospitals is increasing. Difficulties in the management of these children can result in incoordinate decision-making, increased incidence of disagreements with families and a loss of focus on the child's best interests.

### WHAT THIS STUDY ADDS

⇒ A series of statements agreed by a mixed group of professionals at a tertiary children's hospital demonstrate strong clinical consensus about best practice.

### HOW THIS STUDY MIGHT AFFECT RESEARCH, PRACTICE OR POLICY

⇒ The statements can be used by healthcare teams to guide better engagement with families, as well as informing standardised policies to reduce incidence of disagreements and improve decision-making.

for which there is no reasonable hope for cure and from which they will die. The prevalence of children with life-limiting conditions in the UK is increasing.[2]

Decision-making in such cases can be medically, socially and ethically complex. Decisions may be misunderstood and/or contested. Disputes, either between different professional teams or between clinicians and the child's family, are common across clinical settings.[3 4] Although the broad legal framework for making decisions has remained relatively consistent for three decades, society has radically changed. Successive National Health Service reforms have sought to reshape health provision 'in the business of customer service'.[5] While the welfare of the child remains the 'paramount consideration' in law,[6] decisions for and on behalf of children are guided by principles of shared decision-making, where the parent and clinician are on a nominally equal footing.[7 8] This welcomes recognition of partnership notwithstanding,

movements within society assert absolute parental rights to decide what is best for their child's health.[9]

Against this background, advances in medicine mean that increasing numbers of children with complex care needs are surviving for longer and occupying a greater proportion of hospital bed-days.[10] Invariably, such patients fall under numerous medical teams and social agencies, each trying to serve the needs and wishes of the child and their family. Commonly, a lack of dedicated leadership results in incoordinate decision-making, the family confused and the best interests of the child overlooked. Such factors might exacerbate professional–parental disharmony and contribute to complaints and litigation, while compromising the well-being of all concerned parties.[11]

While much has been written on how best to manage conflict in paediatric practice,[12 13] there is less consensus on the best approach to decision-making in children with complex needs. Best practice should help clinicians to compassionately deliver high-quality, medically appropriate clinical care, while successfully collaborating with children, families and colleagues.

## METHODS

We used a modified Delphi process to determine consensus across key professional groups.[14] An initial face-to-face meeting of selected clinical leaders (doctors, nurses, managers) was arranged to identify the key themes underpinning decision-making. Participants received short lectures on essential law and ethics, the evidence base for partnership working and the factors contributing to conflict. There followed three facilitated discussions. In the first discussion, five groups of participants, each containing six to eight professionals from mixed disciplinary and clinical backgrounds, shared experiences among themselves of managing complexity. In the second discussion, groups fed back to one another the problems faced and approaches used in their own practice. Finally, participants developed consensus opinions around key themes, which were then developed after the meeting by CM, ST-U, GB and JF into normative 'statements' around standards and approaches to managing complexity. The resultant themes and statements were finally shared back with the meeting participants who were invited to check them for accuracy. In two cases, additional statements were suggested by participants. These were checked against existing themes and incorporated into the supporting statements.

Two sequential Delphi rounds were used to rank supporting statements within each theme. Participants scored their agreement with each statement using a 5-point Likert scale: 1=strongly disagree, 2=disagree, 3=neutral, 4=agree, 5=strongly agree. Participants also had the option to select 6=don't know. Invitations to take part in the survey were sent by email to multiprofessional staff members across the children's hospital. This survey involved senior managers, all medical consultants, all ward nursing

managers, all clinical nurse specialists (CNS), all psychologists, senior allied healthcare professionals, family liaison team members and the chaplaincy. The email invitation directed each staff member to a REDCap portal where they registered their professional group and level of experience, and were assigned a unique identifying survey number. After 16 weeks the survey was closed. Agreement that a particular statement should be supported and carried forward to the second round was based on the following: 70% of respondents had scored that statement ≥3 with exclusion of those who had indicated 'don't know'.

All participants who participated in the first round of the survey were then emailed an invitation to participate in a second Delphi round. In the second round, participants were asked to again score their agreement with each ranked statement from round 1. In round 2, they were additionally provided with the median score for each statement from all participants, alongside their own score from round 1, to explicitly allow participants to revise their scores on the strength of emerging consensus. The final scores for each statement, from all participants in round 2, were analysed to determine their mean, median and IQRs.

### Patient and public involvement

Patients and the public were not involved in the design or reporting of this research for reasons that are explained in the Limitations section of this article.

### Statistical analysis

Items that were scored 6 (=don't know) were treated as missing data and not included in the statistical analysis. We inspected bar charts in order to assess the most appropriate measure of normality. This provided highly compelling evidence that scores were not normally distributed. Therefore, medians and IQRs (25th–75th percentiles) were used to measure central tendency.

$X^2$ tests were used to determine the strength of evidence for any differences ($p<0.1$) between the median scores of participants who had dropped out between Delphi rounds 1 and 2 compared with those who completed both rounds. We used a p value of 0.1 in order to gauge whether there may have been some weak associations in the data which may have otherwise been dismissed due to the relatively small sample size (rather than as an arbitrary cut point of significance).[15]

Non-parametric K-sample tests were used to compare median responses across different professional backgrounds and years of experience. The large number of statements being compared (69) could lead to small p values purely by chance. To avoid this problem of multiple testing we used the Bonferroni correction to adjust the p value of interest to be 0.0014 (0.1/69). Therefore, no association was deemed noteworthy unless it had a p≤0.0014.

## RESULTS

The initial consensus forming face-to-face meeting proposed 69 normative statements grouped into seven

## Box 1  Themes

**Consensus group themes**
⇒ Standardised approaches to communicating with families.
⇒ Processes for interprofessional communication.
⇒ Processes for shared decision-making in the child's best interests.
⇒ Role of the multidisciplinary team.
⇒ Managing professional–parental disagreement and conflict.
⇒ The role of clinical psychologists.
⇒ Processes to support staff.

key themes (themes summarised in box 1 and full listing of normative statements in table 1).

In the first Delphi round, surveys were sent to 390 professionals and 163 (42%) replied. In the second Delphi round, the 163 professionals who had participated in the first round were surveyed again, and 99 (60%) returned the surveys with all Delphi items scored. Across the 69 Delphi items answered by 99 participants there were only 66 responses (0.9%) of 6=don't know. There was no statistical difference between the responses of those participants who only responded to round 1 of the Delphi survey and those who responded to both rounds 1 and 2.

The professional background of the 99 participants is shown in table 2.

Statistical analysis showed reasonable evidence that professional background affected the median value being assigned for two normative statements. Statement 5 (table 1) was more strongly supported by those from a consultant, clinical nurse specialist or 'other' background, while statement 56 (table 1) was more strongly supported by those from psychology backgrounds.

The years of experience of the participants are shown in table 3.

Statistical analysis showed little evidence that years of experience affected the median values for any of the normative statements.

All normative statements met the a priori threshold (*>70% of respondents scoring each statement greater than or equal to 3*) for proceeding to the second Delphi round. In the second round, while the mean score for each statement ranged from 3.0 (neutral) to 4.9 (strong agreement) (see detail in table 1), there was little evidence of differences across median scores in the manner participants ranked their responses, either across themes or within themes.

## DISCUSSION

The most notable finding in our study was the uniform degree of consensus across a group of hospital professionals on how best to approach decision-making in children with complex care needs. The seven major themes (standardising the approach to communication with families, communication between professionals, shared decision-making, the role of the multidisciplinary team (MDT), managing conflict, the role of clinical psychology and staff well-being)

resonated with colleagues and provided a useful framework for exploring specific principles. While the level of consensus created by the discussions in the workshop and subsequently measured in all themes was striking, there is perhaps some merit for observing those that carried weight and discussing their implications.

The survey highlights the essential importance of good communication both with the family and between professionals. Concepts such as the 'team around the child', a liaison or 'key worker' and a single defined clinical lead are not new but do require some committed resource and intent. All too often parental–professional disharmony arises due to misunderstandings or perceived poor coordination of care. While the survey talks of a 'standardised' approach to communication, this is not intended to imply a 'one size fits all' solution. Each family is unique and good communication is successful because the needs of each participant are individually recognised.[16] There is a delicate balance between structure and intimate spontaneity that must be achieved in practice.

There was a strong agreement that clinicians and parents should work in partnership for the best interests of the child. This is particularly pertinent in the wake of recent court cases[17] where some voices within academic, ethical and legal circles have argued that a shift in authority towards parents in shared decision-making is needed.[18] Participants supported the principle that children have rights that might require independent advocacy. However, while there was a strong consensus that transparency was important, professionals were more ambivalent about the importance of this reflecting their own values and goals (statement 24). Finally, there was a consensus that good care for children might imply a willingness to consider the child in isolation as well as within a family unit, and that one should not automatically assume that children will share their parents' values.

Although children were therefore recognised to be central to professionals' considerations, there was also a strong consensus on the importance of discussing the family's wishes (statements 13, 17, 21 and 22) within a process of shared decision-making (statement 18). While there was an acknowledgement of the importance of seeking parents' views, our survey was not able to adequately explore professionals' insight into the 'lived experience' of a family navigating the distress of caring for a critically ill child in hospital. We suggest that grading of parental behaviour using a 'traffic light system' (statement 51) should not be misinterpreted as a judgemental critique of a family's behaviour but rather as an opportunity for professionals to think about their own assumptions and approaches and a hospital to think about whether extra resources are required to assist mediation. Furthermore, while the effects of conflict on professionals are well reported (statements 61 and 62), we acknowledge that there is less attention to the effect of conflict on the well-being of parents and families.

While there was a clear consensus for most statements (mean scores >/=4) there were some where support was more ambivalent. An example of this was statement 14 that reflected the importance of listening to the child's voice. We

**Table 1** Results after final round of Delphi

| Theme | No | Statement | Mean score | SD | Median | IQR 25 | IQR 75 |
|---|---|---|---|---|---|---|---|
| When communicating with families a standardised approach should be applied to… | 1 | Agreeing to the message communicated to families to avoid giving conflicting ambiguous messages. | 4.4 | 0.8 | 4 | 4 | 5 |
| | 2 | Assessing the families' concerns so as to ensure they are directed to the right person in any discussion. | 4.9 | 0.4 | 5 | 5 | 5 |
| | 3 | Noting discussions. | 4.6 | 0.7 | 5 | 4 | 5 |
| | 4 | Ensuring the preceding antenatal consultations are shared among the treating team in a coordinated manner. | 4.9 | 0.4 | 5 | 5 | 5 |
| | 5 | Acknowledging the contribution of all professionals from consultant to bedside nurse to junior doctor. | 4.6 | 1.1 | 5 | 4 | 5 |
| | 6 | Information sharing among professionals. | 4.8 | 0.5 | 5 | 5 | 5 |
| | 7 | Using 'family held records' to allow families to document meetings, their content and reflections of understanding. | 4.1 | 0.7 | 4 | 4 | 5 |
| When communicating between professionals… | 8 | Processes should be put in place by the hospital trust to strengthen communication between community care providers, hospices and regional hospitals. | 4.6 | 0.6 | 5 | 4 | 5 |
| | 9 | Professionals in the 'team around the child' should be acknowledged and identified early in the child's care pathway. | 4.7 | 0.5 | 5 | 4 | 5 |
| | 10 | A case worker/liaison nurse should be assigned to each complex case to improve coordination across different teams. | 4.7 | 0.6 | 5 | 4 | 5 |
| | 11 | A single clinical lead for the child should be defined across all the involved specialty teams. | 4.7 | 0.5 | 5 | 4 | 5 |
| In the process of shared decision-making in the child's best interests… | 12 | We need to be honest with families about medical uncertainty. | 4.9 | 0.3 | 5 | 5 | 5 |
| | 13 | There should be a holistic assessment of the child and families' wishes, values and goals to inform more complex decisions later. | 4.4 | 0.7 | 5 | 4 | 5 |
| | 14 | We need to, where possible and appropriate, listen to what children have to say. | 3.7 | 1.0 | 4 | 3 | 4 |
| | 15 | Be aware that children may have motivations separate from what they may say (eg, wanting to please their parents). | 4.6 | 1.0 | 5 | 4 | 5 |
| | 16 | Children have rights and these may require independent advocacy even if this means healthcare professionals disagreeing with their parents. | 4.8 | 0.4 | 5 | 5 | 5 |
| | 17 | A holistic assessment of the child and families' wishes, values and goals should include an assessment of their spiritual and religious beliefs. | 4.8 | 0.4 | 5 | 5 | 5 |
| | 18 | We should engage families in 'parallel planning' early and routinely in a child's disease course. | 4.8 | 0.5 | 5 | 5 | 5 |
| | 19 | Professionals need training to understand the legal framework in which they operate. | 4.8 | 0.4 | 5 | 5 | 5 |
| | 20 | Professionals need to be trained in communication techniques. | 4.7 | 0.5 | 5 | 4 | 5 |
| | 21 | A holistic assessment of the child and families' wishes, values and goals should be done in partnership with members of the child's multidisciplinary team (MDT). | 4.5 | 0.7 | 5 | 4 | 5 |
| | 22 | We should try to provide options rather than making closed recommendations. | 4.5 | 0.8 | 5 | 4 | 5 |
| | 23 | Professionals find it distressing when they are unable to fix medical problems and need supporting when this happens. | 4.3 | 0.8 | 4 | 4 | 5 |
| | 24 | We need to transparently share our own values and goals with families and children. | 4.5 | 0.7 | 5 | 4 | 5 |

**Table 1** Continued

| Theme | No | Statement | Mean score | SD | Median | IQR 25 | IQR 75 |
|---|---|---|---|---|---|---|---|
| The multidisciplinary team (MDT) should… | 25 | Be well supported through administrative assistance with preparation and note taking. | 4.4 | 0.6 | 4 | 4 | 5 |
| | 26 | Have outcomes recorded in a consistent and transparent fashion. | 4.3 | 0.8 | 4 | 4 | 5 |
| | 27 | Be recognised in job plans, given that repeated attendance at multidisciplinary team (MDT) meetings by recognised key professionals is onerous and takes time. | 4.6 | 0.6 | 5 | 4 | 5 |
| | 28 | Be held in an appropriate physical environment to enable clarity of discussion. | 4.8 | 0.4 | 5 | 5 | 5 |
| | 29 | Be attended by the wider team including hospital, community and hospice representatives. | 3.6 | 1.0 | 4 | 3 | 4 |
| | 30 | Ensure that appropriate weight is given to all expressed views. | 3.0 | 0.8 | 3 | 3 | 3 |
| | 31 | Be chaired by professionals who are trained in chairing such meetings. | 4.7 | 0.5 | 5 | 4 | 5 |
| | 32 | Only take place when attempts have been made to understand the child's values and goals. | 4.4 | 0.7 | 4 | 4 | 5 |
| | 33 | Be chaired by a professional outside the child's primary clinical team. | 4.1 | 0.7 | 4 | 4 | 5 |
| When managing professional–parental disagreement or conflict… | 34 | Families should be given realistic honestly held opinions. | 4.3 | 0.7 | 4 | 4 | 5 |
| | 35 | We need to better recognise and support mental health issues in families. | 4.3 | 0.7 | 4 | 4 | 5 |
| | 36 | We need to prevent the 'threat response' whereby professionals adopt behaviours to avoid contact with families/each other. | 4.8 | 0.4 | 5 | 5 | 5 |
| | 37 | The organisation should recognise that (in a rights and consumer-based society) any framework that is put in place to improve decision-making may not negate conflict and/or complaint. | 4.7 | 0.8 | 5 | 4 | 5 |
| | 38 | Professional–parental disagreement takes resources and time from other patients and so should be an issue of the highest priority for the organisation. | 4.1 | 0.8 | 4 | 4 | 5 |
| | 39 | The organisation has a responsibility to convey the challenges of decision-making, in the context of patient complexity, to the wider society (local community, NHS leaders, national bodies). | 3.5 | 1.1 | 4 | 3 | 4 |
| | 40 | Families and professionals should receive advice and support on the benefits and risks of social media use. | 4.0 | 1.1 | 4 | 4 | 4 |
| | 41 | Standardised information should be provided to families explaining how the decision-making process works. | 3.6 | 0.8 | 4 | 3 | 4 |
| | 42 | In rare circumstances parental–behavioural contracts are an important tool in addressing disruptive parental behaviour. | 3.7 | 0.8 | 4 | 3 | 4 |
| | 43 | A standardised pathway for decision-making should set out processes for where disagreement arises. | 4.0 | 1.1 | 4 | 4 | 4 |
| | 44 | The Clinical Ethics Advisory Group (CEAG) is an important resource and should be included in any standardised pathway. | 3.8 | 1.3 | 4 | 3 | 4 |
| | 45 | There is a need nationally to standardise processes by which second opinions are sought (when?, who?, how?, with parental engagement?). | 4.1 | 1.1 | 4 | 4 | 5 |
| | 46 | External second opinions should be sought from a national peer-reviewed specialty multidisciplinary team (MDT) where these are available. | 4.3 | 0.6 | 4 | 4 | 5 |
| | 47 | The ward round handover is an important opportunity to identify evolving issues. | 4.7 | 0.5 | 5 | 4 | 5 |

Continued

**Table 1** Continued

| Theme | No | Statement | Mean score | SD | Median | IQR 25 | IQR 75 |
|---|---|---|---|---|---|---|---|
| | 48 | National peer-reviewed multidisciplinary teams (MDTs) should be convened in specialities where they do not exist. | 4.3 | 1.0 | 4 | 4 | 5 |
| | 49 | Gaining an external second opinion is important and should be included in any standardised pathway. | 4.4 | 0.6 | 4 | 4 | 5 |
| | 50 | Gaining a local second opinion is important and should be included in any standardised pathway. | 4.3 | 0.8 | 4 | 4 | 5 |
| | 51 | A 'traffic light system' whereby a family's behaviour is graded allows earlier identification of conflict and should be included in a standardised pathway to reduce conflict. | 4.4 | 1.0 | 4 | 4 | 5 |
| Clinical psychologists should… | 52 | Be integrated within all clinical teams. | 4.3 | 1.0 | 4 | 4 | 5 |
| | 53 | Support the decision-making process. | 4.5 | 1.0 | 5 | 4 | 5 |
| | 54 | Support training in communication techniques. | 4.2 | 1.0 | 4 | 4 | 5 |
| | 55 | Routinely explore family goals and values and explicitly share such information with the multidisciplinary team (MDT). | 4.4 | 0.9 | 4 | 4 | 5 |
| | 56 | Have their role better explained to families. | 4.4 | 0.9 | 4 | 4 | 5 |
| | 57 | Support training in chairing complex multidisciplinary team (MDT) meetings. | 4.2 | 1.0 | 4 | 4 | 5 |
| | 58 | Be involved from the outset in all complex decision-making discussions. | 4.4 | 1.0 | 5 | 4 | 5 |
| In supporting staff… | 59 | Teams need to adopt process where they can come together to discuss challenging cases. | 4.7 | 0.5 | 5 | 4 | 5 |
| | 60 | Teams need to adopt process where individual colleagues are supported. | 4.8 | 0.4 | 5 | 5 | 5 |
| | 61 | The organisation should recognise that parental–professional disagreement (conflict) results in poor morale and staff attrition. | 4.8 | 0.5 | 5 | 5 | 5 |
| | 62 | The organisation should recognise the physical, mental and reputational harms done to professionals in extreme cases of professional–parental disharmony. | 4.8 | 0.4 | 5 | 5 | 5 |
| | 63 | The organisation should recognise that nursing colleagues are particularly vulnerable due to the requirement for them to be at the bedside 24 hours/day. | 4.9 | 0.4 | 5 | 5 | 5 |
| | 64 | The organisation should recognise that HCPs (Healthcare Professionals) ' intrinsic desire to 'do the right thing' through leading in complex cases often over-rides regard for their personal well-being, and increases their vulnerability to experiencing moral distress, compassion fatigue and burnout. | 4.2 | 1.0 | 4 | 4 | 5 |
| | 65 | Professionals should take decisions as a team seeking quoracy (consensus within the core team) wherever possible. | 4.6 | 0.6 | 5 | 4 | 5 |
| | 66 | Staff have a responsibility themselves to access available support options. | 4.2 | 0.7 | 4 | 4 | 5 |
| | 67 | Cross-specialty case forums (such as Schwartz rounds) are helpful for different teams to come together to discuss challenging scenarios. While not part of a decision-making pathway they should be available for staff aftercare. | 3.8 | 0.8 | 4 | 3 | 4 |
| | 68 | The organisation should provide better opportunities for 1:1 and group peer support. | 4.1 | 0.8 | 4 | 4 | 5 |
| | 69 | The organisation should do more to signpost colleagues to available resources. | 4.8 | 0.4 | 5 | 5 | 5 |

HCP, healthcare professional; NHS, National Health Service.

**Table 2** Participant professional background

| Professional background | Participants (n) |
|---|---|
| Consultant | 46 |
| Senior nurse/ward manager | 16 |
| Clinical nurse specialist | 25 |
| Psychology | 5 |
| Allied Health Professional | 3 |
| Family support service | 2 |
| Chaplain | 1 |
| Unknown (did not answer) | 1 |
| Total | 99 |

interpret this apparent ambivalence as professionals taking a pragmatic view in that children with complex needs often have cognitive and communication deficits, or are ventilated on intensive care, which makes knowing the views of the child challenging. That said, an alternative interpretation might be that some professionals are not appropriately inclusive. We would support future studies that look to how the perspectives of parents and children are represented. Engagement of parents and, especially, children may require research processes, such as deliberative methods, that are less demanding and more resource intense than a Delphi process.[19 20]

There was a strong agreement about the importance and structure of the MDT although there was a predominant view that such a meeting need not be chaired by a professional outside the child's primary clinical team (statement 33). Interestingly, there was some ambivalence expressed regarding the inclusivity of these meetings with regard to the attendance of the wider team and the giving of weight to all views (statement 29 or 30). This might reflect a pragmatic view around the logistics of holding such meetings in a timely fashion, or an adherence to intraprofessional considerations of hierarchy and expertise, and where ultimately professionals regard decision-making to lie. In our view, MDT meetings with or without families form the vital platform on which clinical decisions are taken. They should involve all key professionals. Yet it should be acknowledged that they therefore require much planning, should be recognised in job plans and be granted sufficient resources.

When things go wrong, complaint and conflict too often arise. These are challenging for both families and members

**Table 3** Participant years of experience

| Years of experience | Participants (n) |
|---|---|
| < 2 | 11 |
| 2–5 | 21 |
| 5–10 | 23 |
| >10 | 43 |
| Unknown (did not answer) | 1 |
| Total | 99 |

of the clinical team. There was a tacit acknowledgement that, despite best intentions and optimal circumstances, conflict will occasionally still arise. Organisations should give conflict resolution the highest priority and consider standardised approaches to clinical ethics consultation and external second opinions. Referral for external second opinions has been a consistent recommendation for reducing conflict in many previous publications.[12 21 22] However, in our study, while there was a strong consensus for such processes to be standardised at a national level (statement 45), there was slightly more ambivalence around these being introduced as a component of an automatic conflict resolution pathway at a local level (statements 49 and 50). Participants in our study were also supportive of involving national peer-reviewed MDTs where these were available (statement 46) or establishing such bodies where they were not (statement 48).

There was a consensus for clinical psychology support to be integrated within clinical teams, and for them to take an active role in exploring family goals and values and supporting complex decision-making discussions. Finally, there was a strong agreement around the importance of supporting staff, and that in situations of professional–parental disharmony, poor morale and staff attrition result. The deleterious impact on clinicians of difficult relationships and challenging dilemmas in practice has been previously noted.[23–25] A primary focus on individual resilience may miss the importance of investment in institution-wide structures.[26] Healthcare professionals' intrinsic desire to 'do the right thing' through leading in complex cases may override their regard for their personal well-being, and increase their vulnerability to experiencing moral distress, compassion fatigue and burnout.

### Limitations

Importantly, we did not consider the views of either parents or children in this Delphi survey. This is because the survey was conceived as a quality improvement exercise that sought to share best practice between professionals in response to specific institutional challenges. While the exercise would have been improved with the inclusion of parental and children's views, a comprehensive approach would involve a widespread consultation, probably using a more interactive and engaging technique than a Delphi ranking. Unfortunately, such an expanded project was beyond our scope and resources. Further consensus work will need to ensure that the statements included here provide a solid basis for a model for partnership between hospital staff, parents and children, as well as the range of agencies that are also involved in complex care within the community: for example, hospice providers, social work and third sector agencies.

Given the high degree of consensus shown it is worth considering if this consensus was genuine. Our statistical analysis showed that dropping out of the Delphi was not due to being a statistical outlier, suggesting the process did not discourage dissenters from taking part. Nevertheless, 74.5% of those approached did not engage with the process at all.

Although the level of engagement seems broadly comparable with similar Delphi studies[27 28] (and accounting for the fact that non-engagement is a problem common to many types of research), we cannot discount the idea that a proportion of these staff did not value the aims of the study.

## CONCLUSION

The number of children with complex care needs in our hospitals is increasing. Our study of a large mixed group of healthcare professionals in a tertiary children's hospital demonstrates a strong consensus across seven key themes towards developing best practice in decision-making in children with complex care needs: standardising the approach to communication with families, communication between professionals, shared decision-making, the role of the MDT, managing conflict, the role of clinical psychology and staff well-being. Within each of these themes there are notable elements of practice that healthcare teams might engage with to better support the patients and families in their care. In some situations, a comprehensive adoption of a standardised approach might mitigate against professional–parental disharmony.

**Contributors** The project was conceived by JF, ST-U, CM and GB and they all contributed to the face-to-face meeting and developed the statements. JF and CM ran the Delphi survey with MB giving specialist advice on developing the REDCap database, running the survey and interpreting the data. SI gave statistics advice and conducted the statistical analysis. GB led the writing of first and subsequent drafts of the research article. All other authors contributed to revisions and intellectual content. GB is responsible for the overall content as guarantor.

**Funding** GB's research was funded in whole, or in part, by the Wellcome Trust (grant number: 209841/Z/17/Z).

**Disclaimer** The funders had no role in the preparation of this manuscript or the decision to submit for publication.

**Competing interests** None declared.

**Patient and public involvement** Patients and/or the public were not involved in the design, or conduct, or reporting or dissemination plans of this research.

**Patient consent for publication** Not applicable.

**Ethics approval** The study was formally registered with the University Hospitals Quality Improvement 'hub' as a service development project, so formal ethics approval was not required.

**Provenance and peer review** Not commissioned; externally peer reviewed.

**Data availability statement** Data are available upon reasonable request. All data sets are available upon reasonable request to the corresponding author.

**ORCID iD**
Giles Birchley http://orcid.org/0000-0002-2973-2163

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
