## [Reviewer comments · BMJ Paediatrics Open]

ARTICLE DETAILS

TITLE (PROVISIONAL)	Factors affecting decision making in children with complex care needs: a consensus approach to develop best practice in a UK children's hospital
AUTHORS	Birchley, Giles Thomas-Unsworth, Sadie Mellor, Charlotte Baquedano, Mai Ingle, Susanne Fraser, James

VERSION 1 – REVIEW

REVIEWER	Reviewer name: Ms. Sarah Barclay Institution and Country: Medical Mediation Foundation, United Kingdom of Great Britain and Northern Ireland Competing interests: None
REVIEW RETURNED	13-Jul-2022

GENERAL COMMENTS	This is a valuable piece of work which demonstrates consensus about best practice in five key areas of decision-making for children with complex care needs. The authors state clearly at the beginning of the paper and in the limitations sections that patients and families have not been involved in this research but they do not say why and what led them to make the decision not to involve them. I think some additional explanation might help to mitigate the somewhat critical flavour which emerges in three of the statements in particular- 51, 61 and 62. Statement 51 refers to "A 'traffic light system' whereby a family's behaviour is graded allows earlier identification of conflict and should be included in a standardised pathway to reduce conflict" The invitation to health professionals to "grade" a family's "behaviour" feels judgmental and could result in assumptions being made about that family which could result in them being labelled as, for example, a "difficult family." The wording of this statement perhaps missed an opportunity which might have supported clinicians to think differently about their own approaches and assumptions in the discussions which took place during the research process. Statement 61 states that "the organisation should recognise that parental-professional disagreement results in poor morale and staff attrition." Statement 62 refers to "the physical, mental and reputational harm done to health professionals in extreme cases of professional-parental disharmony" While both these statements clearly highlight important impacts and consequences of professional-parent disagreement, I would have liked to see a corresponding statement which reflected an organisation's recognition that such disharmony may also result in poor morale and physical and mental harm for parents and families. If the purpose of this research was to explore the consensus around best practice in decision-making for these challenging and complex cases, it might have been helpful to include statements which allowed participants to reflect on parent/family perspectives alongside those of professionals. A more detailed explanation of why these were not included and why parents/families were not invited
--

	to participate would help to mitigate the risk of unintentional bias in what is otherwise an insightful piece of research.
--	--

REVIEWER	Reviewer name: Dr. Catherine Skellern Institution and Country: United Kingdom of Great Britain and Northern Ireland Competing interests: None
REVIEW RETURNED	03-Jul-2022

GENERAL COMMENTS	This is an excellent study, providing clear directions about these difficult issues, which you have recognised, are increasing in tertiary paediatric hospitals. Your study methodology was well done and the interpretation/discussion of results puts the results into a useful context. The paper overall is well written and worthy of publication. CEAG (page 12 line 28) needs to be defined.
--

REVIEWER	Reviewer name: Dr. Pankaj Garg Institution and Country: South Western Sydney Local Health District, Australia Competing interests: None
REVIEW RETURNED	13-Jul-2022

GENERAL COMMENTS	Reviewers' comments to authors Factors affecting decision making in children with complex care needs: a consensus approach to develop best practice in a UK children's hospital This paper describes Delphi process on understanding the best practices for management of children with complex care needs. The paper is well written and is a useful summary of themes that are generally well known: standardised approaches to communicating with families; processes for inter-professional communication; processes for shared decision-making in the child's best interests; role of the multi-disciplinary team; managing professional-parental disagreement and conflict, and; the role of clinical psychologists. The ongoing challenge is how we deliver the above effectively and in what integrated care frameworks. I have following suggestion for improving the paper:  • Consensus for support of a statement following the close of the survey was calculated after excluding scores for 'don't know', using threshold of "70% of respondents scored that statement ≥ 3". However, this poses question as to many of these responses were neutral and whether this affected true consensus for given statement? • For the second Delphi round, would the revelation of the group median score for each statement contribute to any bias upon the participants responses in Round 2? • Was it possible to use a statistical test of variance to determine the variability of the group of scores? (rather than bar charts). • Why was $p < 0.1$ used as the threshold of statistical significance rather than < 0.05 for the Chi-squared tests regarding the differences of median scores between participants who did not go on to complete the second round versus those participants who completed both rounds.
--

	 • How did the proportions of the different professional disciplines in the sample compare to the original group invited to the Delphi Round 1? Is there an under-representation of allied health professionals? • Was there possibly a bias or misleading content in statement 5, that excluded the acknowledgement or mention of professional backgrounds other than those listed in the statement: 'consultant', 'bedside nurse', 'junior doctor'. Which professional backgrounds made up 'other' background? • What contributed to the relatively weak agreement for upholding the voice of the child in Statement 14? This could be a critical issue. • Is there any comment regarding the discussion in Statement 29, that community and hospice teams are relatively less regarded as relevant? Acknowledgement given to fact that not all complex care cases involve hospice teams, however, complex care for a child often does involve many community providers outside of hospital. Or perhaps, I am misreading this statement 29, with possible typo.  • Would future directions also consider community healthcare, hospice providers, external social/support agencies, as key participants, in addition to parents in children, in this consensus process? (Without this, effective care may remain a utopian dream!) • Is there any comment regarding Statement 30, the ambiguity (3 = neutral) about considering all views expressed in an MDT meeting. Limited to MDT amongst professionals, or with family? In the context of ethically difficult cases e.g. futility of treatment, end of life care? Would suggest expanding on acronyms not already introduced in the text (and less known outside NHS) e.g. CEAG in statement 44. Typos: statement 29, statement 57, line 14-15 of discussion (page 15).
--	--

VERSION 1 – AUTHOR RESPONSE

Dear Editors and peer reviewers,

Many thanks for arranging for the peer review of our article. The reviewers' comments are very constructive and insightful, and we have enjoyed engaging with them. We have carefully considered the reviewers comments and made changes wherever they were clearly indicated.

All reviewers made good points, and we have addressed almost all of these in the revised draft.

Responding to these, particularly in adding extra detail about the statistics and method, have nudged the word count slightly above the 2500 word-limit of the first draft. We trust that this is

acceptable, and you will find the revised paper stronger.

The change tracked copy of the article is included as a supplementary file, with a clean copy provided as the main document for review. The table below is a detailed response that quotes the reviewers comments and responds directly to each point in turn.

Reviewer Comment to the author Author's Response

Editor in Chief Discussion para 3, 2nd

sentence "This is particularly

pertinent in the wake of the

Charlie Gard case" amend to

"This is particularly pertinent

in the wake of recent court

cases"

This sentence has been amended as you suggest.

Associate

Editor

This is a very important issue

and I commend the authors

on embarking on this project

exploring factors affecting

decision making in children

with complex care needs.

The reviewers have raised

some important issues and I

invite the authors to respond

to these comments.

Thank-you for this comment. We respond to the

reviewers below

This is an excellent study, providing clear directions about these difficult issues, which you have recognised, are increasing in tertiary paediatric hospitals. Your study methodology was well done and the interpretation/discussion of results puts the results into a useful context. The paper overall is well written and worthy of publication.

Reviewer: 1 Thank-you for this comment.

Dr. Catherine

Skellern

CEAG (page 12 line 28) needs to be defined.

An explanation of this acronym has been added.

This is a valuable piece of work which demonstrates consensus about best practice in five key areas of decision-making for children with complex care needs.

The authors state clearly at the beginning of the paper

and in the limitations sections that patients and families have not been involved in this research but they do not say why and what led them to make the decision not to involve them.

I think some additional explanation might help to mitigate the somewhat critical flavour which emerges in three of the statements in particular- 51, 61 and 62.

Thank-you for this comment:

We have added some explanation in the limitations to explain why families were not involved in this stage of our research. Rather than repeat ourselves, we have signposted the reader to this explanation in the methodology section at the beginning of the paper.

In addition, we respond to the critique of the statements you have identified below.

Statement 51 refers to "A 'traffic light system' whereby a family's behaviour is graded allows earlier

identification of conflict and should be included in a standardised pathway to reduce conflict” The invitation to health professionals to “grade” a family’s “behaviour” feels judgmental and could result in assumptions being made about that family which could result in them being labelled as, for example, a “difficult family.” The wording of this statement perhaps missed an opportunity which might have supported clinicians to think differently about their own approaches and assumptions in the discussions which took place during the research process.

Reviewer 2:

Ms. Sarah

Barclay,

Medical

Mediation

Foundation

Statement 61 states that

“the organisation should

Thank-you, these comments raise an important point and we have acknowledged these criticisms in the discussion on p14. As we explain, In our view there was adequate attention to the importance of seeking parental viewpoints, but potentially inadequate acknowledgement of the ‘lived experience’ of parents, i.e. the statements perhaps reduced parents to problems to be managed. We suggest that these gaps are evidence that points toward an expanded process that solicits family views in future.

Page 14 para 4, we add:

‘While there was an acknowledgement of the importance of seeking parents’ views, our survey was not able to adequately explore professionals’ insight into the ‘lived experience’ of a family navigating the distress of caring for a critically ill child in hospital. For example, grading of parental behaviour using a ‘traffic light system’ (Statement 51) should not be misinterpreted as a judgmental critique of a family’s behaviour but rather as an opportunity for professionals to think about their own assumptions and approaches, and a hospital to think about whether extra resources are

requires to assist mediation. Furthermore, while the effects of conflict upon professionals are well recognised that parentalprofessional disagreement results in poor morale and staff attrition.” Statement 62

refers to “the physical, mental and reputational harm done to health professionals in extreme cases of professionalparental disharmony” While both these statements clearly highlight important impacts and consequences of professional-parent disagreement, I would have liked to see a corresponding statement which reflected an organisation’s recognition that such disharmony may also result in poor morale and physical and mental harm for parents and families.

reported (statements 61 and 62), we acknowledge that there is less research of conflict upon the physical and mental wellbeing of parents and families’

If the purpose of this research was to explore the consensus around best practice in decision-making for these challenging and complex cases, it might have been helpful to include statements which allowed participants to reflect on parent/family perspectives alongside those of professionals. A more detailed explanation of why these were not included and why parents/families were not invited to participate would help to mitigate the risk of unintentional bias in what is otherwise an insightful piece of research.

Thank-you. We think these points are now covered with the changes we have made to the methods, limitations and discussion sections.

Reviewer: 3

Dr. Pankaj

Garg, South

Western

Sydney Local

Health District

This paper describes Delphi process on understanding the best practices for management of children with complex care needs.

The paper is well written and is a useful summary of

Many thanks for your comments, which we respond to below.

themes that are generally well known: standardised approaches to communicating with families; processes for interprofessional communication; processes for shared decision-making in the child's best interests; role of the multi-disciplinary team; managing professionalparental disagreement and conflict, and; the role of clinical psychologists. The ongoing challenge is how we deliver the above effectively and in what integrated care frameworks. I have

following suggestion for

improving the paper:

- Consensus for

support of a statement

following the close of the

survey was calculated after

excluding scores for 'don't

know', using threshold of

"70% of respondents scored

that statement ≥ 3 ".

However, this poses question

as to many of these

responses were neutral and

whether this affected true

consensus for given

statement?

Thank you. We acknowledge that our use of the

term 'consensus' may have caused confusion

here. We have utilised an approach adopted in

other Delphi studies whereby, excluding 'don't

know', a threshold of greater/equal of 3 (where 3

is neutral) to support which statements go

through to the final round.

We added this sentence to Pg 5 para 2 to

hopefully add clarification:

'Agreement that a particular statement should be

supported and carried forward to the second

round was based upon the following: 70% of respondents had scored that statement ≥ 3 with exclusion of those who had indicated 'don't know'.

- For the second Delphi round, would the revelation of the group median score for each statement contribute to any bias upon the participants responses in Round 2?

The Delphi method we followed specifies that group median score is shared with participants on the second round. The intention is that participants revise their rankings with the aim of group consensus in mind.

We added the text on p5 para 3:

“...to explicitly allow participants to revise scores on the strength of emerging consensus” to make it clear that the information was given for an explicit effect.

- Was it possible to use a statistical test of variance to determine the variability of the group of scores? (rather than bar charts).

It would be possible, but the aim was to assess normality of the variables with a view to assessing whether it was most appropriate to present means or medians. For most of the variables, the bar charts provided highly compelling evidence for non-normality, so there seemed no need to delve further into the distributions.

We have altered the text on p5 para 5 to read:

“We inspected bar charts in order to assess the most appropriate measure of normality.. This provided highly compelling evidence that scores were not normally distributed.”

- Why was $p < 0.1$ used as the threshold of statistical significance rather than < 0.05 for the Chi-squared tests regarding the differences of median scores between participants who did not go on to complete the second round versus those participants who completed both rounds.

Using a p-value of 0.05 to determine

‘significance’ is an arbitrary cutpoint (see DaveySmith and Sterne, BMJ 2001:

<https://www.bmj.com/content/322/7280/226.1>).

We prefer to couch conclusions in terms of the strength of evidence provided by the data, rather than as either 'significant' or not. We used a p of 0.1 in order to gauge whether there may have been some weak associations in the data which may have otherwise been dismissed due to the relatively small sample size.

We have added the following text to the Statistical analysis section on page 6:

“We used a p of 0.1 in order to gauge whether there may have been some weak associations in the data which may have otherwise been dismissed due to the relatively small sample size, rather than as an arbitrary cut point of significance” – we reference the paper by DaveySmith and Sterne we have suggested.

- How did the proportions of the different professional disciplines in the sample compare to the original group invited to the Delphi Round 1? Is there an under-representation of allied health professionals?

Thank you. If by the “original group” you are referring to the face to face meeting, the proportions are slightly different. The face-to face meeting involved a wide range of senior

professionals with backgrounds in medicine, nursing, AHP, management, palliative care, psychology, patient liaison and chaplaincy. (We believed it was necessary to have a 'broad church' to advise on the key themes). Inevitably, the subsequent survey to all staff gained responses in round 1 from professional groups (such as AHP) in different %s to those who had attended the original meeting since each exists in varying numbers across the hospital (clinical staff> nursing> AHPs> management etc.)

We used standard trust group email lists to recruit, and do not have the precise employment data from the hospital of how many of each type of professional was employed at the time the survey was conducted. However we believe that professional representation in the survey responses broadly reflects proportions of staff from different disciplines involved in this work across the organisation. Given the absence of precise data we did not feel that adding a statement addressing your question would add to the clarity of the statistical analysis in this case.

Do please clarify if we have misunderstood your question.

- Was there possibly a

bias or misleading content in statement 5, that excluded the acknowledgement or mention of professional backgrounds other than those listed in the statement: 'consultant', 'bedside nurse', 'junior doctor'. Which professional backgrounds made up 'other' background?

Thank you. It is correct that statement 5 refers to "all" professional backgrounds. We do not consider this to present a serious lack of clarity, and it would be difficult to write a succinct statement that mentioned every professional background at work in a tertiary hospital who might interface with the patient and their family

After consideration we suggest that it is unnecessary to address this in the discussion.

- What contributed to the relatively weak agreement for upholding the voice of the child in Statement 14? This could be a critical issue.

Thank you. This is an interesting observation.

We have added a paragraph to page 15 para 1:

‘While there was clear consensus for most statements (mean scores 4, 5) there were some where support was more ambivalent. An example of this was Statement 14 ‘the need, where possible and appropriate, to listen to what children have to say’. We interpreted this response as colleagues taking a pragmatic view in that children with complex needs often have cognitive and communication deficits, or are ventilated on intensive care, that make knowing the views of the child challenging. That said, an alternative interpretation might be that some professionals are not appropriately inclusive. We would support future studies that look to how the perspectives of parents and children are represented. Engagement of parents and especially children may require research, such as deliberative methods, that are less demanding and more resource intensive than a Delphi process.

- Is there any comment regarding the discussion in Statement 29, that community and hospice teams are relatively less regarded as relevant?

Acknowledgement given to fact that not all complex care cases involve hospice teams, however, complex care for a child often does involve many community providers outside of hospital. Or perhaps, I am misreading this statement 29, with possible typo.

We think that a typo had affected the way this statement is interpreted . Corrected it refers to “the wider team including hospital, community and hospice representatives”, and so does not ‘single out’ hospice and community colleagues specifically.

However, we agree that the fact that this statement and statement 30 scored relatively lower merits discussion.

We have therefore iterated the relevant section on page 15 para 2:

‘This might reflect a pragmatic view around the logistics of holding such meetings in a timely fashion, or an adherence to intra-professional considerations of hierarchy , expertise, and where ultimately professionals regard decision making to lie’

- Would future directions also consider community healthcare, hospice providers, external social/support agencies, as key participants, in addition to parents in children, in this consensus process? (Without this, effective care may remain a utopian dream!)

This is a good point. We have acknowledged this point in the Limitations section page 16, para 3, adding the statement that further research will include parents and children...“as well as the range of agencies that are involved in complex care within the community: for example hospice providers, social work and third sector agencies”

- Is there any comment regarding Statement 30, the ambiguity (3 = neutral) about considering all views expressed in an MDT meeting. Limited to MDT amongst professionals, or with family? In the context of ethically difficult cases e.g.

futility of treatment, end of
life care?

Thank you. Please see earlier comment regarding
statements 29 and 30.

We use the term multidisciplinary team to
describe the group of professionals (from a
variety of clinical disciplines) who are directly
engaged with the patient and their parents and
who come together to make decisions , with the
family, regarding recommended treatment
options. They would form the 'team around the
child' . It is common practice for parents to be
invited to an MDT meeting.

Would suggest expanding on
acronyms not already
introduced in the text (and
less known outside NHS) e.g.
CEAG in statement 44.

I have expanded the acronym CEAG where it
occurs – I could find no other examples in the
text.

Typos: statement 29,
statement 57, line 14-15 of
discussion (page 15).

Thank-you, I have corrected these typos